# Telemedicine in Oral and Maxillo-Facial Surgery: An Effective Alternative in Post COVID-19 Pandemic

**DOI:** 10.3390/ijerph17207365

**Published:** 2020-10-09

**Authors:** Ida Barca, Daniela Novembre, Elio Giofrè, Davide Caruso, Raffaella Cordaro, Elvis Kallaverja, Francesco Ferragina, Maria Giulia Cristofaro

**Affiliations:** Maxillo-Facial Surgery Unit, Department of Experimental and Clinical Medicine, “Magna Graecia” University, Viale Europa, 88100 Catanzaro, Italy; barca.ida@gmail.com (I.B.); daniela.novembre@gmail.com (D.N.); eliogiofre@gmail.com (E.G.); carusodavide81@gmail.com (D.C.); raf.fy.86@hotmail.it (R.C.); reikey2003@yahoo.com (E.K.); francesco.ferragina92@gmail.com (F.F.)

**Keywords:** coronavirus, maxillofacial surgery, telemedicine, telephone interview, intraoral photography

## Abstract

The aim of this work was to demonstrate the advantages of using telemedicine (TM) in the management of the outpatients with maxillofacial surgical pathologies during the COVID-19 pandemic. The study was conducted at the MaxilloFacial Surgery Unit of “Magna Graecia” University of Catanzaro, on two different groups of patients: a group of follow-up patients (A_1_: patients in oncological follow-up after surgical treatment performed before the COVID-19 pandemic; A_2_: suffering from chronic lesions such as precancerous lesions), and a group B of patients with first urgent visits (B_1_: patients with suspected oncological pathology; B_2_: patients with suspected urgent disease such as medication-related osteonecrosis of the jaws (MRONJ), odontogenic abscesses, temporomandibular joint (TMJ) dislocation, etc.). Participation in the study required possession of a smartphone with Internet access, e-mail and the use of a messaging service (WhatsApp or Telegram) to send photos and messages; completion by the patient of a COVID-19 screening questionnaire; submission of a satisfaction questionnaire by the doctors and patients. A total of 90 patients were included in this study. A high percentage of satisfaction emerged from the analysis of the satisfaction questionnaires of both patients and doctors.TM thus represents an excellent opportunity to improve accessibility to oncological and non-management activities, reducing the risk of Covid-19 dissemination and should be promoted and implemented in the post-pandemic era.

## 1. Introduction

To minimize the interruption of health services during the COVID-19 pandemic, especially for urgent outpatient visits and oncological follow-up, TM meetings have been implemented and adopted by many academic medical centres, becoming an important tool in this period of limited face-to-face interaction [1,2]. TM can be defined as the set of telematic technologies used to provide the patient with health care services—diagnosis, monitoring, therapy—at distance. It is therefore a distance delivery of health services. For the World Health Organization, TM literally means “remote healing” and indicates the use of technologies, information and communication to improve the health outcomes of patients, increasing access to medical care and information. What qualifies the TM is therefore the prefix “Tele”, that is the distance from the health worker who provides the service, regardless of the technology with which the contact is made. Before the COVID-19 pandemic, TM, although available for years, was gradually entering medical practice [3] and in any case used above all in chronic patients to offer home remote monitoring solutions that guaranteed continuity of care by preventing emergency situations and avoiding hospitalizations. Usage was somewhat limited due to numerous regulatory restrictions.

In Italy as early as July 2012, the Superior Health Council approved the Italian guidelines on telemedicine, and according to experts of the time, technological innovation would have been able to reorganize healthcare through innovative care models, which would have facilitated access to services on the national territory. With the stipulation of the Pact for Digital Healthcare in 2016, we continued to talk about TM as a means of providing healthcare services and the need to implement tele-visits, tele-consultations and tele-cooperation healthcare to achieve important system changes, as well as to achieve big savings. But what developments and benefits have there been in the meantime? What real benefits has the use of this “new” technology brought to doctors and patients? Unfortunately, it is not easy to find data on the effective implementation of the Pact and an emergency like COVID-19 was necessary to return to talking about the usefulness of TM, although it is certain that communication technologies, such as smartphones, tablets and laptops, have supported the rapid development of telemedicine as a new concept of health care to provide remote assistance [4,5,6]. The most significant advantages of TM are consultation in real time and archiving and forwarding of data; however, TM is not without drawbacks: in fact, the first concern when it comes to telemedicine is always its accessibility because a availability of stable connections in remote areas is often problematic and increases the concern for poorer patients with limited access to the Internet [7]. Another obstacle is the impossibility of carrying out an adequate clinical examination at a distance, but the main problem is the selection of patients to be visited in the clinic and those to be subjected to remote visits [8,9]. From the experience acquired by the authors during the COVID-19 pandemic, it emerged that TM is indicated above all in the follow-up of cancer patients, because they are immunosuppressed for previous chemo/radiation treatments and therefore subject to greater risk of contagion, and chronic as well as urgent visits for suspected malignancy [10]. Certainly, the COVID-19 emergency made us talk about the usefulness of TM as it made it essential to resort to tele-medical practices to avoid overcrowding in hospitals and the risk of a massive spread of the virus also in hospitals. More precisely, this study describes its advantages in the management of outpatient maxillofacial surgical pathologies during the COVID-19 pandemic.

## 2. Materials and Methods

The study was conducted from 29 February to 30 April 2020 at the Oral and Maxillofacial Surgery Outpatient of “Magna Graecia” University of Catanzaro, Italy. Our hospital was designed to be a regional reference Centre for COVID-19 because represents the only centre that has an intensive care unit with the ability to treat patients with severe forms of infection that require extracorporeal oxygenation (ECMO) techniques and with high specialization in the management of severe forms of acute respiratory distress syndrome (ARDS). A clinical management protocol for both suspected and confirmed cases was adopted and shared, in order to support the centres that require patients affected by related COVID-19 ARDS forms to be centralized at our COVID-19 Centre. The study protocol, submitted to the Ethical Committee of the Magna Graecia University of Catanzaro and with code reference number 146 of 21 May 2020, was conducted in accordance with the “Ethical Principles for Medical Research Involving Human Subjects” described in the Helsinki Declaration. All patients gave their informed consent to their participation in the study and the storage of their data.

### 2.1. Study Protocol

All patients spontaneously joined the study and gave their consent to the data processing. The inclusion criteria of the study were the following: (1) possession of a smartphone with a video camera and microphone with Internet access and an institutional email address in order to send imaging reports and photos to a device; (2) installation of the WhatsApp or Telegram applications that use end-to-end encryption for telephone messaging; (3) a completed COVID-19 screening questionnaire (Table 1); (4) a satisfaction questionnaire completed by the doctor and the patient (Table 2). The exclusion criteria were: (1) suspected or confirmed COVID-19 positive patient (patients with claimed symptoms of COVID-19); (2) patients with objective difficulties in participating in the study (inability to connect to the Internet, the presence of diseases that conditioned the connection); (3) age under 18. The study included two different groups of patients: follow-up patients (group A) and patients with a first urgent visit (group B).

The A group is divided into two subgroups:A_1_ patients in oncological follow-up after surgical treatment performed before the COVID-19 pandemic;A_2_ patients suffering from chronic lesions such as precancerous lesions, MRONJ;

The B group is divided into two subgroups:B_1_ patients with suspected oncological pathologyB_2_ patients with suspected urgent disease (MRONJ, odontogenic abscesses, TMJ dislocation)

Before being subjected to a remote visit, all patients were contacted by telephone for adherence to the protocol and to explain the limits of the method related to not performing a physical examination in person and submitting the COVID-19 screening questionnaire; the authorization to use telemedicine was obtained via recorded video or signature on a specific consent form and sent on the institutional email.

For the scheduled video call, we were asked to dress professionally, minimize ambient sounds, make sure the physical environment was appropriate, hold the webcam directly in front of the face, keep all the imaging documentation related to the disease available. An alternative to “face-to-face” evaluation was offered in a timely manner in cases where the limitation of an incomplete physical examination could increase the risk of an incorrect diagnosis. If patients requested it, an electronic medical prescription could be sent to them at any time.

### 2.2. A Group

For the patients of subgroup A_1_ the evaluation was carried out through CD visualization, clinical photos taken by the patient or a family member, evaluation of symptoms. For the chronic lesions in follow-up (subgroup A_2_) we considered any signs of deterioration through the modification of the tissue morphological characteristics, in the case of MRONJ the possible variation of the stage and the presence of infection signs were assessed.

### 2.3. B Group

For the first visit procedures, for both subgroups, the evaluation of the clinical picture was conducted both through the video call that the evaluation of the clinical photos sent (necessary in the case of pathology of the oral cavity or laterocervical region) and through the display of symptoms.

### 2.4. Data Collection

The data collected included: patient demographics, patient photos (head, neck, oral cavity), time consultation, doctor and patient feedback. The photos had to be taken by the patient himself or with the help of a direct relative using the camera of his smartphone to acquire only the following images: (1) a photo of the face and/or neck; (2) a photo of the oral cavity; (3) a photo of the maximum buccal opening with a visible millimeter ruler. This data had to be sent to the examiner’s smartphone using a messaging service like WhatsApp application that uses end-to-end encryption, so the communication between the sender’s phone and the recipient’s phone is secure. Furthermore, the phone to which the photos were sent was only accessible to a small group of professionals through the use of specific authentication credentials.

The photos and imaging reports were recorded in the patient’s medical record (electronic or physical), consultations were not recorded. Particular attention has been paid to the conservation, transmission and use of patient data, in compliance with the ethical and legal responsibilities of confidentiality and professional secrecy, in full compliance with the General Data Protection Regulation (GDPR) in force since 4 May 2020. Patients in both group A and B, for whom a biopsy was required, underwent a preoperational antimicrobial mouth rinse has been performed to reduce the number of oral microbes. However, as instructed by the Guideline for the Diagnosis and Treatment of Novel Coronavirus Pneumonia (the 5th edition) released by the National Health Commission of the People’s Republic of China, chlorhexidine, which is commonly used as mouth rinse in oral practice, may not be effective to kill 2019-nCoV. Since 2019-nCoV is vulnerable to oxidation, pre-procedural mouth rinse containing oxidative agents such as 1% hydrogen peroxide or 0.2% povidone has been used, for the purpose of reducing the salivary load of oral microbes, including potential 2019-nCoV carriage.

Healthcare staff used the following personal protective equipment (PPE): N95 or FFP2 mask, eye protection, fluid-resistant gown, and surgical gloves. The surgical therapeutic protocol adopted for surgical procedure included the use of scalpel over monopolar cautery for mucosal or skin incision and bipolar cautery on lower power setting for haemostasis and absorbable sutures. The patients were discharged after a period of observation in an individual room with home cold and medical therapy.

### 2.5. Data Analysis

Descriptive statistical analyses were performed on the recorded data using absolute frequencies and percentages for categorical data. The recorded data were subjected to descriptive statistical analysis. Both the central tendency indices (such as mean and median) and the absolute and relative frequencies were calculated using the GraphPad program (GraphPadCompany, San Diego, CA, USA).

## 3. Results

### 3.1. Demographic Characteristics and Pathology of the Study Population.

A total of 90 patients were included in this study from 29 February to 30 April 2020 (Table 3). In the study 54 patients were males (60%) and 36 females (40%) with a man to female ratio of 1.5:1. The range age was of overall 17 to 95 with a mean age of 62.15 years. Of all patients, 37 were resident in the province of Catanzaro and 53 were from other provinces of Calabria with an average distance of 103 km between home and hospital.

### 3.2. Results A Group

The A group included 63 follow-up patients: subgroup A_1_ was formed by 54 patients (86%) with oncological pathology after surgical treatment performed before the COVID-19 pandemic, subgroup A_2_ was formed by nine patients (14%) with chronic lesions of which six were precancerous and three MRONJ (Table 4). In the subgroup A_1_, for 44 patients (81%) the video consultation combined with imaging (computed tomography (CT), magnetic resonance imaging (MRI), ultrasound neck and salivary lodges) and the clinical photos did not show signs of loco-regional and remote recurrence; for three patients (5%) of subgroup A_1_ undergoing recent surgery for oral cancer, the examination of the photos required an advanced dressing (Figure 1).

In seven cases the video consultation with imaging information and the symptoms that emerged from the interview (pain, burning and difficulty in chewing and speaking) were sufficient to determine the need of in-person consultation and in five cases the need for urgent surgery (Figure 2). In these cases, the surgery was booked remotely and the final patient assessment was conducted on the day of their operation, reducing the time of surgery and the number of hospital visits. Subsequently, the histological report confirmed the recovery of the disease. All group A patients, who underwent biopsy examination, sent photos on the third and seventh day post-surgery to check the locoregional conditions. On average, each consultation required 15 ± 5 min.

### 3.3. Results B Group

The B Group included 27 patients with “first urgent visit”: subgroup B_1_ was formed by 12 (44.4%) patients of which three were for suspected skin cancer; eight for oral cancer and one for submandibular gland cancer; subgroup B_2_ was formed by 15 patients (55.5%) of which 7 for MRONJ, two for TMJ dislocation, three for odontogenic abscesses, three for suspicion of acute salivary gland pathology (Table 4).

According to the examination of the clinical photos and conversations with the patients of subgroup B_1_, an outpatient visit and simultaneous biopsy examination (two for the skin and four for the oral cavity) were necessary for six patients (Figure 3 and Figure 4). For six patients of subgroup B_2_, an outpatient visit was necessary in two cases for the reduction of the TMJ dislocation, in one for the drainage of a dental abscess (Figure 5) and in three for dressing and purulent collection drainage of MRONJ (Figure 6). All B group patients who underwent biopsy examination sent photos on the third and seventh day post-surgery to check the locoregional conditions. On average, each consultation required 15 ± 5 min.

### 3.4. Analysis of Satisfaction Questionnaires

All patients contacted joined the study and were able to submit the requested photos and read and signed the administered satisfaction questionnaire. The analysis of the data obtained from the questionnaire administered to collect patient feedback (Figure 7) has shown that:(1)73% of patients found easy to participate in the consultation, 20% medium and 7% difficult.(2)78% currently preferred telemedicine, 12% indifferent and 10% face-to-face consultation(3)80% of patients chose video-telephone consultation rather than telephone consultation because they were able to see the doctor’s face and because it was easier to describe the symptoms.(4)92% of patients would recommend video consultation to others.

The analysis of the data obtained from the questionnaire administered to collect the feedback from the doctors of maxillofacial surgery unit involved in the telemedicine service (Figure 8) has shown that:97% of doctors found it easy to participate in the consultation, 2% medium and 1% difficult.the method proved to be 92% safethe resolution of the video image to evaluate facial asymmetries, the presence of swellings, bone exposures, skin and mucous membrane changes due to the presence of suspected lesion was 89% satisfactory98% of doctors would recommend video consultation to other colleagues.

## 4. Discussion

The new coronavirus pandemic has dramatically affected health organizations around the world and the effect on health systems, their resources and clinical services has been profound. Because the new coronavirus is highly contagious, there has been an increasingly urgent need to devise and identify new patterns of care delivery to avoid “face-to-face” consultations between doctors and patient and to reduce the risk of transmission. In particular, maxillofacial surgeons, specialists much more at risk than others for close contact with the mouth and upper airways of patients during diagnosis and treatment, have been forced to find alternative ways of assistance in compliance with the new restrictions that the Italian government has had to adopt for people, requiring them to stay at home and to limit their social life [11].

In Italy, TM has played a decisive role in reducing the risk of spreading COVID-19; in this period of health emergency, through videoconference and/or photography we were able to:evaluate the surgical sites (possible presence of visible masses and/or local recurrences, evident lymphadenopathies, etc.),discuss with the patient about the progress of oncological pathology,view the cancer surveillance imaging,allow the patient to ask questions and clear up doubts,involve other specialists in the video-conference, because, as is known, the treatment of head-neck oncological pathology requires a multi-disciplinary approach (maxillofacial surgeon, radiotherapist, oncologist, nutritionist, etc.).

We considered it essential to explain why we did not perform the physical visit, obtaining authorization from the patient to use remote assistance via recorded video, written message or signature of a specific consent form [12,13]. The study showed that through an adequate anamnesis in the video call and the careful evaluation of the patient’s photographic book it was able to satisfactorily check the patients in follow-up for neoplastic and chronic lesions (precancerous lesions, MRONJ) putting comparing the photos received with those included in the photographic archive of the structure [14,15,16]. The 15 patients who requested the first visit were managed with a remote consultation that considered as differentiable pathologies the three cases of sialadenitis, two cases of MRONJ and two odontogenic abscesses, to which we prescribed cold [17], antibiotic and anti-inflammatory medical therapy and a subsequent remote revaluation at 7 and 15 days.

It should be emphasized that in patients with suspicion of recurrence of neoplastic disease, telemedicine allowed us to recognize in 13% lesions with a high suspicion of malignancy that required immediate surgical intervention confirmed with histological examination in 9%. The high percentage of satisfaction that emerged from the analysis of the satisfaction questionnaires administered to the patient shows that telemedicine was well accepted especially by the patients residing in areas rather far from the structure, who had to travel a long way and above all be exposed to other patients who could be carriers of COVID-19 and other more common infectious agents. Above contact, although remotely, has positively influenced the patient’s state of well-being, motivation and sense of security specially in this uncertain period. Of all patients, 37 were resident in the province of Catanzaro and 53 were from other provinces of Calabria with an average distance of 103 km between home and hospital. In some of these countries, severe restrictions had also been adopted to strengthen the containment of COVID-19. Without digitization and an online conversation, the imposed social isolation would not have allowed for the follow-up of their clinical development. The limitations to the method were the impossibility of carrying out a complete clinical evaluation and a data connection that is not always reliable. This can be overcome by better documentation, shared decision-making and pragmatic management, because the COVID-19 pandemic is an opportunity for healthcare organizations to reassess the crucial role played by telemedicine in the clinical mission, through the optimization of the software and procedures, together with the development of telesurgery due to improvements in 5G data transmission [18]. If implemented, the telemedicine platform will bring a variety of potential benefits for both patients and healthcare organizations [3,19]. In the field of maxillofacial surgery, it will be possible to think of a wider use as in the traumatology which, as is known, does not represent an emergency condition but a deferrable urgency [20]. Through a telematics meeting of a multidisciplinary team, it is possible to plan the treatment, through the visualization of the radiological images, and program the surgical intervention to minimize the duration of the hospital stay beyond that health spending.

## 5. Conclusions

Right now, telemedicine represents an excellent opportunity to improve accessibility to oncological and non-oncological treatments, compensating for the inevitable setbacks and opening a window for the future. The experience acquired by the authors during the COVID-19 pandemic suggests that telemedicine is indicated above all in the follow-up of cancer patients, because they are immunosuppressed for previous chemo/radiation treatments and therefore subject to greater risk of contagion, in chronic as well as urgent visits for suspected malignancy and in patients who live far from a hospital. Social distancing measures have made it difficult to access health facilities without while eliminating the risk of contagion between health personnel and patients. The government limitations implemented, the patient’s awareness of being constantly monitored for his pathology through teleconsultation, contributed both to increasing patient compliance and to establishing a more effective doctor-patient relationship. The post-pandemic role of telemedicine depends on regulatory solutions which, in our opinion, will always have to take into consideration the empathic doctor-patient relationship, which would risk, pushing technology extremely, to a dehumanization of care. Although the priority today is to limit the spread of COVID-19, also we must think of an optimal and safe recovery of non-urgent medical activities that have been delayed due to the emergency and not to increase the spread of the infection within the healthcare facilities. In fact, the social distancing measures will make access to medical clinics difficult and will not eliminate the risk of contagion between health personnel and patients.

## Figures and Tables

**Figure 1 ijerph-17-07365-f001:**
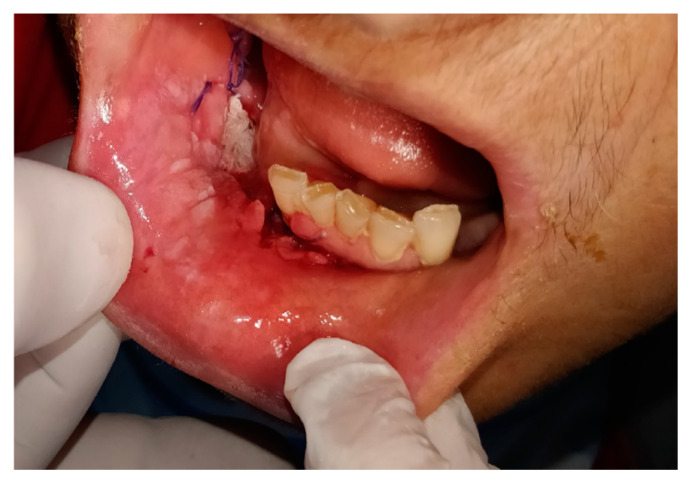
Advanced dressing of the surgical site in patient treated for the cancer of the right alveolar crest during COVID-19 pandemic.

**Figure 2 ijerph-17-07365-f002:**
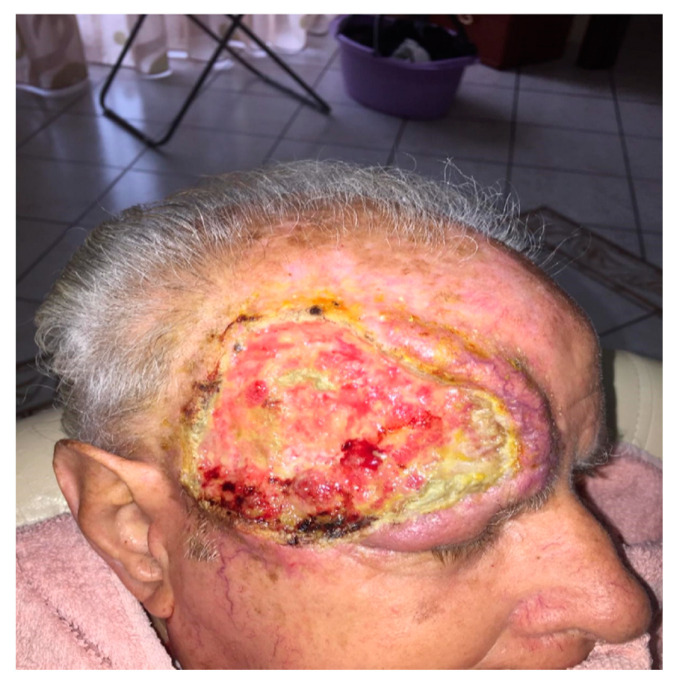
Patient in cancer follow-up with bloody ulcerated necrotic vegetative lesion located in the frontal-temporal skin.

**Figure 3 ijerph-17-07365-f003:**
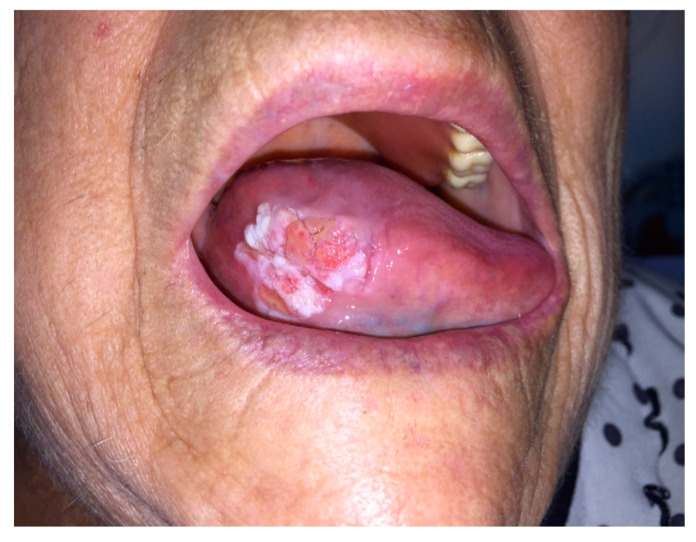
First visit in patient with right lingual margin neoformation.

**Figure 4 ijerph-17-07365-f004:**
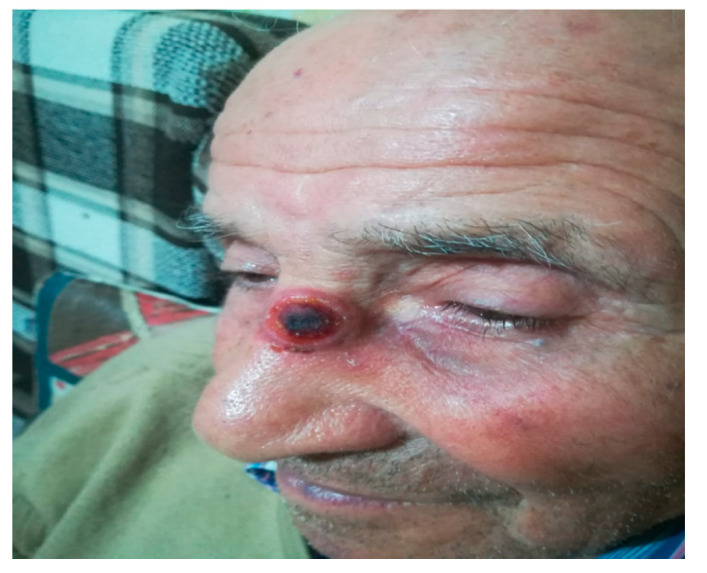
First visit in patient with ulcerated skin nasal lesion.

**Figure 5 ijerph-17-07365-f005:**
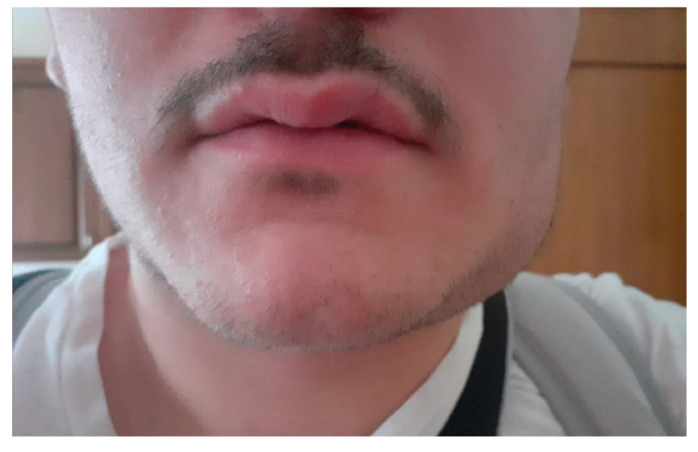
First visit to patient with odontogenic abscess left mandibular angle.

**Figure 6 ijerph-17-07365-f006:**
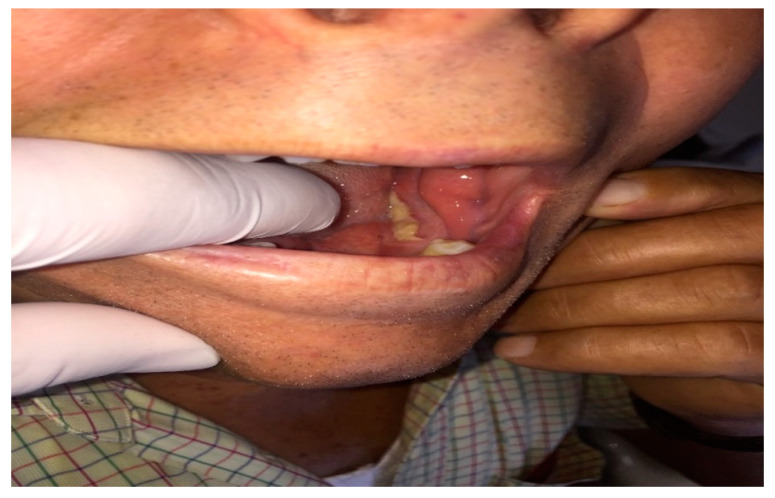
First visit in patient with bone exposure in oral cavity and history of bisphosphonates treatment by several years for prostate cancer.

**Figure 7 ijerph-17-07365-f007:**
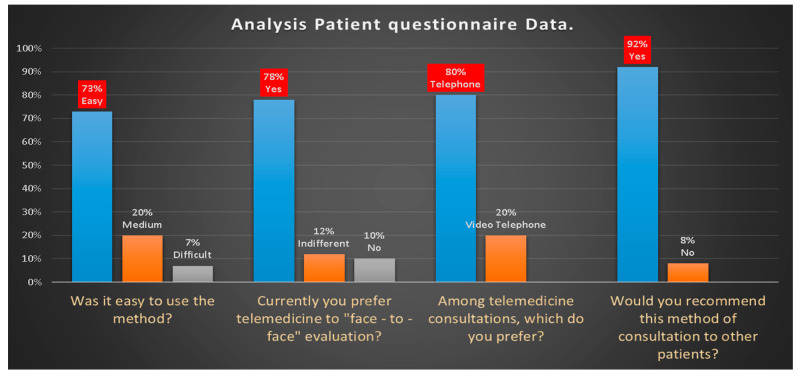
Analysis patient questionnaire data.

**Figure 8 ijerph-17-07365-f008:**
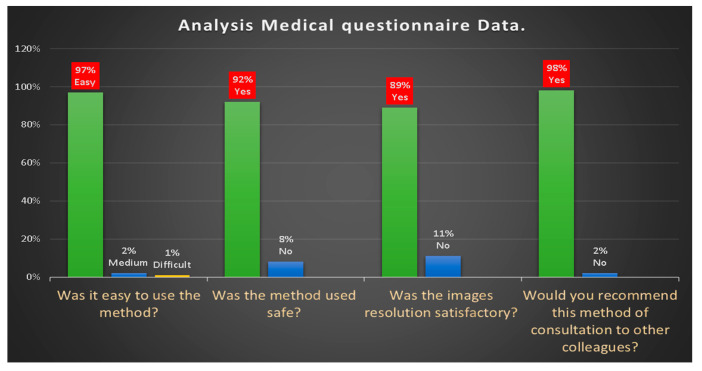
Analysis medical questionnaire data.

**Table 1 ijerph-17-07365-t001:** COVID-19 screening questionnaire.

1. Do you have fever or have you experienced fever within the past 14 days?	YES or NO
2. Have you experienced a recent onset of respiratory problems, such as a cough or difficulty in breathing or diarrhea, ageusia, anosmia within the past 14 days?	YES or NO
3. Have you, within the past 14 days, travelled to risk areas or visited a neighborhood with documented 2019-nCoV transmission?	YES or NO
4. Have you come into contact with a patient with confirmed 2019-nCoV infection within the past 14 days?	YES or NO
5. Have you recently participated in any gathering, meetings, or had close contact with many unacquainted people?	YES or NO

**Table 2 ijerph-17-07365-t002:** Satisfaction questionnaire for the patient and the doctor.

**Patient Questionnaire**	
(1) Was the method easy to use?	▯ Easy ▯ Medium ▯ Difficult
(2) Currently you prefer telemedicine to “face-to-face” evaluation?	▯ Yes ▯ Indifferent ▯ No
(3) Among telemedicine consultations, which do you prefer?	▯ Telephone consultation ▯ Video—Telephone consultation
(4) Would you recommend this method of consultation to other patients?	▯ Yes ▯ No
**Medical Questionnaire**	
(1) Was the method easy to use?	▯ Easy ▯ Medium ▯ Difficult
(2) Was the method used safe?	▯ Yes ▯ No
(3) Was the image resolution satisfactory?	▯ Yes ▯ No
(4) Would you recommend this method of consultation to other colleagues?	▯ Yes ▯ No

**Table 3 ijerph-17-07365-t003:** Demographic characteristics of the study population.

Sex	Mean Age	Residence
Male	68.35	Province of Catanzaro 25
		Other provinces 29
Female	66.23	Province of Catanzaro 17
		Other provinces 19

**Table 4 ijerph-17-07365-t004:** Results of the A and B group.

**Group A**
**Pathology**	**Telemedicine**	**Type**	**Outpatient Treatment/Service**
SubGroup A_1_ Neoplasm Follow-up	54 contacts	Head & neck Carcinoma: 54	Clinical control: 7
		Advanced dressing: 3
		Biopsy: 5
SubGroup A_2_Other Control/Follow-Up	9 contacts	Pre-cancerous: 6	
	MRONJ: 3	
**Group B**
**Pathology**	**Telemedicine**	**Type**	**Outpatient Treatment/Service**
SubGroup B_1_Neoplasm 1st visit	12 contacts	Skin Carcinoma: 3	Skin Biopsy: 2
	Mucosa Oral Carcinoma: 8	Oral Biopsy: 4
	Submandibular gland Carcinoma: 1	
SubGroup B_2_Other 1st visit	15 contacts	MRONJ: 7	MRONJ: 3
	TMJ Dislocation: 2	TMJ Dislocation: 2
	Odontogenic abscess: 3	Odontogenic abscess: 1
	Sialoadenitis: 3

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
