# Peer review of "Telemedicine in Oral and Maxillo-Facial Surgery: An Effective Alternative in Post COVID-19 Pandemic"

_ijerph, 2020, doi:10.3390/ijerph17207365_

Round 1

Reviewer 1 Report

Dear Authors,

First of all, I would like to congratulate you on the study, because in this time it is complicated.

It is an original study, in the field of oral surgery, although with certain improvements.

The translation into English would have to be revised.

The introduction is well written, but the main aim "to illustrate" I do not think is appropriate in this manuscript.

Materials and Methods

The study was approved by the Ethical Committee of the Magna Graecia University of Catanzaro, however the code referring to the approval of that project does not appear in the manuscript, generally the projects have an associated code. It is also necessary to cite everything related to informed consent.

On page 4 of 11, I think the order is not quite right, on the one hand in line 101 "Descriptive statistical analyzes ...." should go just before the results section. On the other hand, 3.1 GROUP A appears and just after GROUP B, it is not well understood what it refers to.

More statistical data are missing, like the type of program used, type of more detailed analysis...in some studies they even ask for the databases.

Within GROUP B the paragraphs should also be ordered, because what refers to sections A and B should go at the beginning.

Results

Although we are aware of the difficulty of taking photographs, we do not know which photographs have been taken by professionals or which by patients, because some of them can be improved.

Has the training of the professionals in video consultation been evaluated?

Best regards

Author Response

Dear Reviewer,

First of all, I would like to thank for the time dedicated to the review, the appreciation and the suggestions given which were invaluable in improving the quality of the manuscript to make it suitable for publication.

Below are the changes made based on what is suggested:

Introduction

  • The term "illustrate" has been changed to "describe"

Materials and methods

  • The code that refers to the approval of the study by the Ethics Committee has been included in the study.
  • All patients gave their informed consent to their participation in the study and the storage of their data.
  • Page 4 of 11: the descriptive statistical analysis was moved before the Results and the type of program used was indicated.
  • The paragraphs of groups A and B have been ordered.

Results

  • The photos were taken by patients or a close relative, not by professionals also because, due to restrictions, it was not possible to go to a professional photographer.

Reviewer 2 Report

Firstly, I thank the Editorial Committee for the opportunity to review this manuscript. The study proposed by its authors is interesting and complies with the purposes of the journal. In addition, the topic covered is quite actual and relevant during the current exceptional situation caused by the COVID-19 pandemic.
However, the methodological rigour of the manuscript should be improved for its publication. Consequently, below I suggest several recommendations addressed to improve the quality of the proposed study.

ABSTRACT.
-The objective of the study is not clear, since the authors analyse the COVID-19 contagion and satisfaction perceived by both doctors and patients, apart from describing their telemedicine proposal. In addition, I consider the last sentence included in the objective of the study (“to be promoted and implemented in…”) is not appropriate, since it should be included in the Discussion or Conclusions section.
-Material and Method: it is recommended to include the study design and preserve the anonymity of the study setting.
-Results: There are no results related to the COVID-19 contagion nor satisfaction perceived by both doctors and patients. Acronyms should be defined (MRONJ and ATM).
-Conclusions: they are not related to the objective nor obtained results.
-Keywords: the use of the Medical Subject Headings (MeSH) terms is recommended; new terms may be used if they are terms that have recently appeared and do not yet appear in MeSH.

INTRODUCTION: I consider it should be improved introducing deeply this topic quite relevant. There are only 3 references in the Introduction section. In this sense, it is highly recommended to include appropriate references related with COVID-19 pandemic and its impact on healthcare, telemedicine (definition, modalities of this healthcare attention, patients who can be treated more effectively, advantages and disadvantages), and the current use of telemedicine during this pandemic. The objective should be improved (see above).

MATERIALS AND METHODS: it is recommended to include the study design and more description of the possible population or patients (participants), and study setting, preserving its anonymity. Both screening and satisfaction questionnaire has not been validated. In this sense, I would like to know why it is relevant to know the COVID-19 contagion: Does it imply suspected of confirmed COVID-19 positive patients could not be treated using this telemedicine modality? Why? This modality seems to be the ideal modality of healthcare for these patients… Regarding the questionnaires applied, more description of its characteristic is needed. Table 3 should be included in the Results section. Acronyms should be defined the first time after using them (MRONJ and ATM). Was a virtual platform used to conduct video consultations or patients were only contacted by telephone? It not clear in the text. A more detailed description of healthcare attention related to the remote contact with patients is needed. It is recommended to include the software used to analyse quantitative data.

RESULTS: Which was the response rate of participants? How many patients were excluded following the exclusion criteria? How many patients were contacted? How many patients were suspected of confirmed COVID-19 positive patients? How many alternative “face-to-face” evaluations were offered to the participants? Where are Graphic 1 and 2? Acronyms should be defined the first time after using them (MRONJ, ATM, and TMJ).

DISCUSSION: The first paragraph of this section (line 201 – 219) should be included in the Introduction section. Include exclusion criteria in study limitations.

CONCLUSION: This section should be rewritten since conclusions are not related to the study objectives nor results obtained. Some sentences should be included in the Introduction or Discussion section.

REFERENCES: There are only 15 references. It is recommended to include more references.

Author Response

Dear Reviewer,

First of all, I would like to thank for the time dedicated to the review, the appreciation and the suggestions given which were invaluable in improving the quality of the manuscript to make it suitable for publication.

Below are the changes made based on what is suggested:

Abstract:

  • Objective: It has been completely changed based on what was suggested. The phrase "Telemedicine represents an excellent opportunity to improve accessibility to cancer and non-cancer care, reducing the risk of contagion" has been moved to the Conclusions.
  • Materials and Methods: The Study design was included.
  • Results: the high percentage of satisfaction with the questionnaire perceived by patients and doctors was included.
  • The acronyms have been defined.
  • Table 3 was included in the results.
  • Conclusions: they have been modified based on the objective and results.
  • Keywords: they have been modified using MeSH

Introduction:

  • It has been detailed as suggested; More bibliographical references have been added, the definition of telemedicine, the advantages and disadvantages, the indications to it, before and during the pandemic.

Materials and method:

  • The study design was included (anonymity had already been preserved).
  • Table 3 has been moved to the results.
  • The questionnaires were validated because they were approved by the Ethics Committee to which we submitted them.
  • We wanted to study a homogeneous sample of patients for whom it was not possible, given the restrictions, to undergo a visit or follow-up.
  • As for the questionnaire, it seemed to us very well detailed.
  • Patients were only contacted by telephone.
  • 12% were not satisfied with the remote visit and 10% were indifferent.
  • The software used was included.

Results:

  • All 90 patients contacted joined the study, none of them met exclusion criteria.
  • We have inserted graphs 1 and 2.
  • The acronyms have been defined.

Discussion:

  • The paragraph "from line 201-219" in the Introduction has been moved.
  • The limitations of the study were highlighted in the discussion.

Conclusion:

  • The conclusion has been rewritten.
  • As suggested some sentences have been inserted in the Introduction and Discussion section.

References:

  • Multiple references were included

Reviewer 3 Report

The new coronavirus pandemic has dramatically affected health organizations in the world. Due to its  highly contagious, there has been an increasingly urgent need to devise and identify new patterns of care delivery to avoid "face-to-face" consultation between doctor and patient and to reduce the risk of transmission. The study evaluated the use of telemedicine in the management of oral disease, This study has clinical implications which will promote and implemente the Telemedicine in oral and maxillo-facial surgery in the post-pandemic. 

1.The lack of a period in last sentence of "MATERIALS AND METHODS".

Since February 2020, the new coronavirus pandemic has dramatically affected health organizations in the world and created severe test to the health system.  Becasue its highly contagious, telemedicine   is expected to become a effective alternative in post COVID-19 pandemic. This study evaluated the feasibility that telemedicine represents an excellent opportunity to improve accessibility to cancer and non-cancer care, reducing the risk of contagion, which have important implications for the daily operation of oral and maxillo-facial surgery. In fact, telemedicine  has alos been introduced to our daily operation during the outbreak of COVID-19. Moreover, the study was rational designed, and the writing was clear and concise. 

I thin this paper is worthy of publication in this journal. 

Author Response

Dear Reviewer,

I would like to thank for the time dedicated to the review, the appreciation and the suggestions given which were invaluable in improving the quality of the manuscript to make it suitable for publication.

Round 2

Reviewer 1 Report

Dear authors,

First of all, I would like to thank the changes in the manuscript. As a reviewer I accept the new manuscript.

Kind regards

Author Response

Dear Reviewer,

I would like to thank for your approval.

Reviewer 2 Report

Firstly, I thank the Editorial Committee for the opportunity to review this manuscript again. The authors have followed most of the recommendations proposed, responding point by point to all of them. Consequently, they have highly improved the quality of the manuscript, mainly its methodological rigour. However, the references section need a minor revision, because the last four references are incorrectly enumerated.

Author Response

Dear reviewer,
I would like to thank for the new suggestions.As suggested, the references have been modified.
